# Taming a 'fuzzy beast'? Stakeholder perspectives on Antarctic science-policy knowledge exchange practices in New Zealand

**Natasha Blaize Gardiner**[1,2]*, **Neil Gilbert**[1,2,3], **Daniela Liggett**[1]

1 Gateway Antarctica, Centre for Antarctic Studies and Research, School of Earth & Environment, University of Canterbury, Christchurch, New Zealand, 2 Antarctica New Zealand, Christchurch, New Zealand, 3 Constantia Consulting Limited, Papanui, Christchurch, New Zealand

* Natasha.gardiner@pg.canterbury.ac.nz

**Data Availability Statement:** As per the conditions of this project's Human Ethics approval, the raw data (including de-identified data) are only

## Abstract

Antarctic environmental change is accelerating with significant regional and global consequences making it critically important for Antarctic research knowledge to inform relevant policymaking forums. A key challenge is maximising the utility of evidence in decision-making, to which scholars have responded by shifting away from linear science-policy arrangements towards co-production alternatives. As an Antarctic Treaty Consultative Party (ATCP), New Zealand (NZ) is responsible for facilitating knowledge exchange (KE) among Antarctic science and policy actors at national and international levels. However, at present, we have few metrics for assessing the success of science-policy dialogues. Furthermore, studies on the Antarctic science-policy interface have so far primarily focused on the international perspective. This paper is the first to examine domestic stakeholder perspectives regarding Antarctic KE using NZ as a case study. We report on the findings of two workshops involving over 60 NZ Antarctic stakeholders in 2021 that aimed to explore the various elements of NZ's Antarctic science-policy interface and identify barriers or drivers for success, including future opportunities. Our results indicate that there is a desire to shift away from the current linear approach towards a more collaborative model. To achieve this, stakeholders share an understanding that KE practices need to become more equitable, inclusive and diverse, and that the policy community needs to play a more proactive and leading role. Described as a 'fuzzy beast', the NZ Antarctic science-policy interface is complex. This study contributes to our understanding of Antarctic KE practices by offering new guidance on several key elements that should be considered in any attempts to understand or improve future KE practices in NZ or within the domestic settings of other ATCPs interested in fostering science-policy success.

## Introduction

Antarctica and the Southern Ocean are critical components of the Earth System, and research has indicated that over the coming decades these regions are expected to undergo significant

accessible to the research team. All data requests with respect to this project should be directed to the University of Canterbury Human Research Ethics Committee (human-ethics@canterbury.ac. nz) using the reference HEC 2020/48/LR-PS.

**Funding:** NBG, NG and DL received funding for this research from the Antarctic Science Platform through grant ASP-023-4. The funders had no role in study design, data collection and analysis, decision to publish, or preparation of the manuscript. https://www.antarcticscienceplatform. org.nz/.

**Competing interests:** NBG is a Policy Advisor for Antarctica New Zealand and NG serves on the Antarctica New Zealand Board of Directors, however, this research does not represent the views of any organisation or country. All authors declare no competing interests.

environmental changes due to direct and indirect anthropogenic stressors [1–8]. The acceleration of Antarctic environmental change has foreboding and disproportionate consequences at local, regional and global scales [2,9,10]. It is therefore of critical importance that the knowledge gained through Antarctic research informs relevant policy and decision-making forums [11,12]. Maximising the utility of evidence in decision-making, however, is not a straightforward undertaking [13], which is reflected in the reported persistence of this challenge across diverse environmental governance arenas [14–18].

In the Antarctic context (by 'Antarctic', we refer to the continent, its surrounding ice shelves and the Southern Ocean up to the Polar Front), the states that are active within the governance forums of the Antarctic Treaty System (ATS) are primarily responsible for the production, management, dissemination and use of Antarctic expertise and knowledge. Delivering this mandate requires Antarctic researchers, policymakers and other actors to engage in multi-directional knowledge exchange (KE) practices that mobilize scientific evidence and other types of knowledge in decision-making processes [11]. The Antarctic Treaty Consultative Parties (ATCPs) have repeatedly affirmed their commitment to using the best available scientific evidence to inform and guide Antarctic decision-making [19] and several formal mechanisms exist to facilitate Antarctic KE. Notably, the Scientific Committee on Antarctic Research (SCAR) has a long history of providing credible, relevant and legitimate scientific evidence to inform Antarctic policy forums [20] and the Committee for Environmental Protection (CEP) is required to draw on scientific and technical expertise to formulate advice and recommendations to the Antarctic Treaty Consultative Meetings (ATCM) on issues of environmental concern. Additionally, the 1980 Convention on the Conservation of Antarctic Marine Living Resources (CCAMLR) established a Scientific Committee to provide the best available evidence to the CAMLR Commission on marine conservation issues [21].

Evidence-based decision-making is a core tenet of Antarctic governance, which requires an alignment between the needs and requests of the Antarctic policy community and the evidence syntheses provided [21]. However, asking policymakers to articulate their research needs in a format that is easily actionable by the research community has been identified as a key challenge both within Antarctic science-policy KE [20] and for science-policy interfaces (SPIs) more broadly [18,22]. Further compounding this challenge is a disconnect between the internationally agreed priorities under the ATS and the domestic institutions of the ATCPs including, for example, the domestic science funders who are partly responsible for incentivising and resourcing policy relevant research projects [11,23]. This raises important questions regarding the roles of Antarctic stakeholders at the national level: to what extent do Antarctic science and policy actors co-produce knowledge domestically to address nationally and internationally important Antarctic research and policy priorities? To what extent do Antarctic stakeholders understand how domestic level science-policy KE arrangements affect their ability to meet their international obligations?

New Zealand (NZ) is an active participant within the ATS [24,25] and its Antarctic interests span political, biophysical, economic, scientific, socio-cultural and historical dimensions [25–29]. The recently updated national Antarctic science strategy *Aotearoa New Zealand Antarctic and Southern Ocean Research Directions and Priorities (2021–2030)* [30] outlines NZ's Antarctic research priorities over the next decade and points to links between research areas and relevant policy forums. Through its guiding principles, the strategy requires Antarctic research to "be well connected to end users and support evidence-based decisions" [30:3], and it draws strong links between NZ's Antarctic research outputs and ATS policy bodies, including the ATCM, CEP and CCAMLR. Accordingly, there have been increasing calls for NZ Antarctic researchers to demonstrate to funding providers how the outcomes of their work will deliver to the needs of decision-makers or society [31].

Despite the apparent importance of maintaining strong Antarctic science-policy connections both internationally and within NZ, it is not always clear how Antarctic KE occurs in practice at domestic scales. Studies that investigate the Antarctic SPI have so far primarily focused on the international governance perspective (e.g., see [11,12,21,32–35]), which leaves domestic KE practices largely unexamined by scholars. In the NZ context, only a handful of previous works have sought to address specific dimensions of the Antarctic SPI (e.g., see [36,37]). There have otherwise been no attempts to situate NZ's Antarctic science-policy KE practices within the expansive field of science and policy scholarship, nor has there been an exploration of how NZ practitioners perceive the efficacy of this nexus. The importance of strengthened coordination across domestic institutions and actors has now been identified as critical for the achievement of the environmental protection and conservation objectives set out by the various instruments of the ATS [11,38–40]. Consequently, without further research that explores domestic Antarctic science-policy interactions, potential opportunities to share empirical lessons across the ATCPs and improve Antarctic KE in NZ and elsewhere remain limited.

In this article, we take a step towards addressing the knowledge gaps outlined above by undertaking a critical qualitative analysis to explore the perspectives of Antarctic stakeholders regarding the current state of Antarctic KE practices in NZ. We report on the findings of two stakeholder workshops (convened in NZ in 2021) that address two central research questions: 1) 'How does the Antarctic science-policy interface(s) function in New Zealand?' and, 2) 'What are the drivers and barriers for success?' After describing our conceptual framework and methods used to explore stakeholder perspectives, we focus our discussion on three pertinent themes that we generated through our analysis: 'a desire to move beyond the linear model', 'the inclusivity-exclusivity paradox of an informal social network and small community', and 'a need for strengthened domestic policy leadership'. We focus on these themes as we consider them as being symbolic of how we can advance our current understandings of science-policy arrangements in NZ. Finally, we discuss the implications of our findings in the context of broader science-policy discourse and offer a summary of opportunities (including future research avenues) that could enhance Antarctic KE practice and theory in NZ and beyond.

## Conceptual framework

In a rapidly changing world, efforts are continually on the rise to support the mobilization of expert knowledge within decision-making processes to address urgent socio-ecological challenges [17,41]. Consequently, scholarship that investigates the relationship between science and policy has proliferated [18,42,43], although analytic pursuits aiming to better understand the role of evidence in public policy are not new (e.g., see [44–46]). To guide our analysis, we develop a conceptual framework (Fig 1) that draws insights from this expansive field of science and policy studies and includes ideas that stem from Science and Technology Studies (STS) [47–50].

We use the term 'knowledge exchange' practices (KE or KE practices) to describe the socially and politically constructed interchange that occurs between science and policy actors [53], and which encompasses "all activities and facets of knowledge production, sharing, storage, mobilization, translation and use" [43:203]. A myriad of context-specific variables influence the ways in which KE practices are arranged and mobilised [18] and KE 'success' comes in diverse forms [41,43].

An increasingly hegemonic discourse throughout KE theory and practice is the need to shift away from linear science-policy practices towards co-production alternatives [18,50,54–56]. Consequently, co-production endeavours are rapidly growing [57] and have been

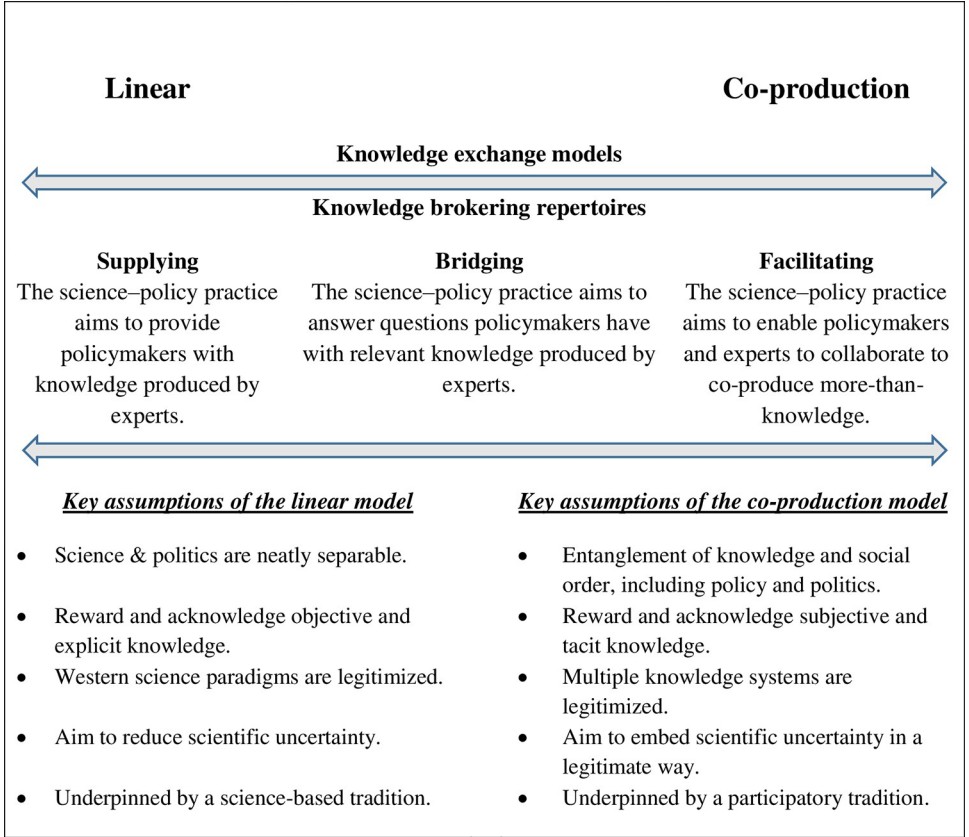

**Fig 1. The knowledge exchange continuum (including KE models, repertoires and assumptions [18,51,52]).**

described as "one of the most important ideas in the theory and practice of knowledge and governance for global sustainability" [58:88]. In large part, this transition reflects a gradual shift in science's role in society and public policy, which has placed different repertoires of knowledge brokering along a continuum of KE arrangements [51]. Whereas science once helped to clarify and make sense of the world around us (i.e., normal science [59]), now science and technology can serve to both solve and create political problems [60]. Thus, claims that science is a neutral or value-free enterprise have been aptly replaced by nuanced descriptions of the entanglement of knowledge and socio-political spheres [50].

Situated at one end of the KE continuum (Fig 1), co-production offers a holistic lens through which to view knowledge production and use, and can be defined as "processes that iteratively bring together diverse groups and their ways of knowing and acting to create new knowledge and practices to transform societal outcomes" [55:322]. Co-production moves beyond science-based approaches towards a more participatory tradition that departs from untenable value-free ideals [60–63]. Underpinned by the facilitating repertoire, co-production gives equal weighting to expert and other types of knowledge, and decisions are based on what is meaningful to decision-makers [see 52]. Co-production outcomes include 'more-than-knowledge' (Fig 1), which refers to knowledge types that are less tangible and not able to be codified in the same way as traditional forms of knowledge, examples include tacit knowledge or 'know how' [18,64,65].

At the other end of the continuum, linear KE practices are underpinned by (post)positivist philosophies wherein science is neatly separable from politics and is objective in nature [14,66]. The associated knowledge brokering repertoires include supplying or bridging,

whereby knowledge production occurs largely in isolation from policy, and the knowledge produced is later delivered in a science-to-policy direction [18]. The key difference is that the bridging repertoire aims to deliver to specific policy needs, whereas the supplying repertoire is focused on the communication of scientific findings irrespective of their relevance to policy problems (e.g., scientists adopt a 'pure scientist' role [67]). Underpinning these technocratic bridging and suppling repertoires are several assumptions (Fig 1) including, for example, a belief that the provision of more timely information, or that the systematic reduction of uncertainty, will lead to improved or more rational decision-making [68]. Although linear KE arrangements are increasingly viewed as unrealistic and epistemologically inadequate for addressing the complex socio-ecological challenges we face [50,62,69,70], they are found to persist across a variety of contexts [18,71,72].

While we situate linear and co-production models as two ends of a continuum in Fig 1, this conceptualisation is limited. We have described two clear-cut extremes that scarcely overlap in their approaches, which we acknowledge is a gross oversimplification. For example, contextual factors may foster linear KE practices, but this does not mean that the individual actors active in these processes necessarily embrace linear assumptions. Likewise, it is possible for co-production to be enacted by individuals or groups that are firmly rooted in linear epistemologies, i.e., there is a distinction between KE activities and practices, and the repertoires used to justify and provide meaning to them [72]. In the context of Antarctic KE in NZ, we do not suggest that one or the other end of the continuum is superior, but we explore the repertoires invoked by Antarctic stakeholders in the context of a broader shift in social science discourse that favours co-production alternatives. While the linear model has come under increasing critique, co-production similarly comes with its warnings. There are risks that require careful consideration prior to embarking upon co-production initiatives [55,57,73–76] and we discuss these later in the context of our results.

## Method

We convened two stakeholder workshops in NZ and analysed the data collected through breakout group and plenary discussions. Workshop 1 involved a 2-hour in-person workshop in the margins of the biennial NZ Antarctic Science Conference (February 2021). The purpose was to bring together a broad range of Antarctic stakeholders to develop a shared understanding of the Antarctic SPI in a NZ context and identify opportunities and barriers for achieving successful science-policy interactions. Workshop 2 involved a 5-hour online workshop in December 2021. The purpose was to bring together Antarctic stakeholders who were not employed in academia, including policymakers, to explore the current state of Antarctic science-policy interactions and identify any potential new or improved ways of working. The second workshop built on the preliminary findings from the first workshop (see S1 File).

### Participants

Workshop 1 was attended by 64 stakeholders. Over half identified as Antarctic researchers ranging from early to late career (Table 1).

Workshop 2 targeted Antarctic stakeholders not primarily engaged in academic pursuits. This included Antarctic policymakers (i.e., policy practitioners that engage with Antarctic research knowledge to inform the development and implementation of Antarctic related policies at the international Antarctic Treaty level and the domestic level) and individuals from a variety of stakeholder entities with varying degrees of interest in Antarctic policy and research. Participants included 24 individuals from 13 stakeholder entities (Table 2), some of whom had participated in Workshop 1. There was a strong bias towards central government entities.

**Table 1. Workshop 1.** Number of participants and their broad affiliation type.

| Primary affiliation | Number of participants |
|---|---|
| University or Crown Research Institute | 39 |
| Government Department or Agency | 16 |
| Regional Council | 2 |
| Consultancy | 1 |
| Other Antarctic stakeholder group | 6 |
| | **64** |

## Breakout groups and data collection

The workshops included a combination of context-setting presentations, breakout group and plenary discussions (S1 File). Written and audio data was collected at both workshops. Written data consisted of the notes recorded by participants during the breakout group discussions. Workshop 1 audio data included full recordings of the breakout group discussions. Workshop 2 audio data included recordings of the plenary discussions. All audio data was transcribed verbatim for analysis.

## Defining KE 'success'

Workshop participants explored the various elements that influence KE practices and how they have an impact on KE 'success'. However, success is not a straightforward notion. Research has shown that successful KE practices are described and evidenced in a variety of ways and are influenced by a breadth of factors (e.g., preconditions or strategies for success) [41,43,77]. Given its conceptual opacity (and due to time constraints), we chose to provide participants with a definition for 'success' to help frame their thinking on this notion and to avoid lengthy discussions on the topic (noting that this is worthy of further scholarly attention). We considered KE practices as successful when they comprised "social processes which encompass relations between scientists and other actors in the policy process, and which allow for exchanges, co-evolution, and joint construction of knowledge with the aim of enriching

**Table 2. Workshop 2.** Number of participants and their primary affiliation.

| Primary affiliation: | No. of participants: |
|---|---|
| Antarctica New Zealand | 6 |
| Christchurch Antarctic Office, Christchurch City Council | 1 |
| Climate Change Commission | 1 |
| Department of Conservation (DOC) | 1 |
| Environmental Protection Agency (EPA) | 1 |
| Land Information New Zealand (LINZ) | 1 |
| Maritime New Zealand | 2 |
| Ministry of Business, Innovation and Employment (MBIE) | 3 |
| Ministry for the Environment (MfE) | 1 |
| Ministry of Foreign Affairs and Trade (MFAT) | 4 |
| Ministry for Primary Industries (MPI) | 1 |
| New Zealand Customs Service | 1 |
| Parliamentary Commissioner for the Environment (PCE) | 1 |
| | **24** |

decision-making" [66:807]. We expand on our use of this definition in the Supplementary Materials.

## Data analysis

The analysis was conducted using Braun & Clarke's [78] reflexive thematic analysis approach (reflexive TA), wherein the analytic process was primarily informed by a social constructionist paradigm [79,80]. Reflexive TA emphasises the active role of the researcher and treats subjectivity as an essential tool that serves to enrich the analytic process. Given the experiential knowledge and embedded professional practice of the authors of this article in the NZ Antarctic science-policy space, reflexive TA presented a suitable approach.

The analytic process was highly iterative, involving six steps: (1) data familiarisation, (2) systematic coding, (3) initial theme generation, (4) theme development and review, (5) theme refinement and (6) writing up. Coding was undertaken by the first author both manually and using NVivo 12. In the first instance, an inductive approach was taken to code all of the relevant data extracts, but as familiarity with dataset grew alongside the development of the conceptual framework, deductive codes were also generated. This resulted in a combination of data-driven semantic codes and researcher-driven latent codes [78,81].

Code maps were used to draw connections between the codes, following which six initial themes were generated. We adopt Braun & Clarke's definition of a theme as "a pattern of shared meaning organised around a central organising concept" [78:77]. Following the initial theme generation, we returned to both the coded extracts and the whole dataset to re-calibrate our thematic ideas against the raw data (i.e., to check our themes actually stood up). This led to further iterations, including reworking some of the original codes and recreating the coding maps. We finally arrived at three themes–one deductive/latent and two inductive/semantic– that meaningfully addressed the research question. These themes are explored more fully in the results and discussion sections and are substantiated by a range of quotes from workshop participants, which are presented verbatim and may hence include the grammatical or orthographical errors contained in the original transcripts.

## Limitations

Given the complex nature of SPIs, we provided participants at both workshops with additional prompts and definitions to help frame their thinking on the issue (see S1 File for further detail). Likewise, we provided specific elements or categories for participants to discuss during both workshops to facilitate conversations. We also used topics that arose in workshop 1 to stimulate discussions in workshop 2. It is therefore possible that the early identification and provision of broad categories and definitions, as well as the use of the first workshop's findings for the second, was too deterministic and narrowed the workshop discussions somewhat. To address these limitations and add to the sincerity and rigour of our work, we spent significant time reflecting on how these predilections may have influenced the data collected and our interactions with the data, concluding that the choices we made were unavoidable in the context of facilitating rich discussions about a complex topic in a short space of time.

The relatively unconstrained and free-flowing nature of the workshop discussions meant that it was often difficult to determine what level of Antarctic 'policy', 'policymaking' or 'policymakers' the participants referred to as the terms were frequently used loosely or in a very broad sense. For this reason, we did not attempt to differentiate between specific Antarctic policy contexts in NZ, but we note that several participants pointed to differences between domestic KE practices in the context of CCAMLR as opposed to the ATCM and CEP. Exploring such differences is beyond the scope of this paper but merits future investigation.

## Results

The workshops facilitated diverse discussions about Antarctic KE practices in New Zealand. The first workshop revealed that participants perceived science-policy interactions as comprising a range of elements spanning various scales (e.g., influential factors at individual, organisational, institutional, government and international levels). The interconnectedness of these elements were emphasised throughout all workshop discussions and there was a resounding struggle across the breakout groups regarding the challenge of disentangling the various components of KE. Such complexity was succinctly summarised by a workshop 2 stakeholder: *"the science-policy interface is a very fuzzy beast and it's across a lot of different interests"* (W2).

### A desire to move beyond the linear model

Our analysis highlights a tendency towards the supplying and bridging repertoires of knowledge brokering and exchange, which serve to reinforce a linear approach to both enacting and giving meaning to KE. While these findings are similar to recent studies that also explore KE models in practice (e.g., see [18,72,82]), we found that Antarctic stakeholders share a desire to shift the current KE arrangement beyond the linear model, towards an approach that embodies co-production characteristics. As one stakeholder expressed, the current model exacerbates challenges that are commonplace for science-policy interactions, such as ensuring that KE is relevant and useful:

> *"Typically the science or research we do is not co-designed with policymakers and therefore may not be fit for purpose and then I think that potentially leads to some miscommunication about what do policymakers want, what do the scientists produce, and how do those two things align or not"* (W2).

In line with the linear model, expectations regarding the roles played by KE actors were primarily described in transactional terms, with an emphasis on the need to 'translate' information so that it could flow from knowledge 'producers' to 'end-users'. Linked to this was the idea that scientists and policymakers lack a 'common language', which saw the breakout groups pointing to a need to either improve science communication, more clearly articulate the policy needs, or develop boundary expertise to provide more effective translational services. For example, one group wrote: *"[we] need to train people to translate between science and policy–[this] should be treated as its own valid specialty"* (W1). While diverse types of boundary work and knowledge brokerage have proven their worth across a range of contexts [22,53,83], the breakout groups predominantly conjured linear imaginaries when conceptualising boundary spanning in the context of Antarctic KE. For instance, one researcher invoked the bridging repertoire when speaking positively about a recent shift towards more transdisciplinary research approaches:

> *"What we're doing more and more is actually talking to end users and stakeholders and saying 'well what is it you need?' So you identify the problem, you co-design an approach and then you work together on taking the evidence and the science and being able to use it in a way that addresses the problem"* (W1).

Despite a desire to move beyond the linear model, the semantics used by stakeholders suggested an internalization or bias towards the supplying and bridging repertoires of KE. Unlike the aims of the facilitating repertoire underpinning the co-production model (see Fig 1), stakeholders described the goals of 'co-design' or 'co-production' in terms of tailoring scientific

products or answers to specific policy-relevant questions and ensuring that such packages of information were delivered in the right way at the right time.

To move beyond the linear model, stakeholders shared a view that maintaining 'two-way' dialogues between science and policy communities would be critical for building trust, cooperation and mutual understandings, and emphasized a need for more frequent face-to-face interactions between science and policy actors. A variety of opportunities to increase engagement were identified including targeted workshops, seminars or engagement forums, increasing the transparency of current engagement pathways and developing capacity building initiatives. Importantly though, the quantity and quality of KE practices were not seen as analogous. Instead, there was a concern that scientists and policymakers may continue to misunderstand each other even when engaging in collaborative practices. In order to embrace the facilitating repertoire, one stakeholder alluded to co-production approaches requiring stakeholders to embrace new ways of working, including a willingness to portray empathy and open mindedness within KE interactions:

> *"You have to understand people's incentives, and their uncertainties and their fears. If you don't understand that then you'll never get through. Doesn't matter how loud you shout. You have to understand the other side. They have their practical considerations, you know, you just have to acknowledge that everybody has their difficulties"* (W1).

Importantly, science-policy arrangements are not the sole products of the beliefs, values and actions of the individuals within them and the breakout groups spoke at length about the many systemic and institutional factors that create barriers at the SPI. In particular, the incentive structures embedded within the international science system and to some extent, the NZ science system, were perceived as a major barrier for scientists to engage in co-production practices. The 'publish or perish' model, for example, was seen to drive scientists to remain in their disciplinary silos and allocate time and effort to purely scientific projects, rather than engaging in time-consuming and often complex policy-relevant work. Despite efforts to incentivize impact-driven research – both globally and in NZ – one researcher voiced concerns related to the pressures that the global impact agenda places on the research community:

> *"Spending all that time in engaging with communication or policy. . . I think that we're a little bit blind to the level of effort it actually requires to do all this stuff. You know we throw this stuff out as, 'oh we should be doing this and we should be doing that'. . . But you should also be doing your science because you need to be internationally recognized. I think there's too much being thrown at you as a scientist"* (W1).

Additionally, stakeholders did not always see impact-driven Antarctic science projects in NZ making genuine or meaningful attempts to engage with policy. Instead, policy engagement was generally described as occurring at the beginning or the end of the research process, with scientific products eventually being retrospectively mapped on to policy needs or requirements as an afterthought.

## The inclusivity-exclusivity paradox of an informal social network and small community

NZ's Antarctic science and policy communities are small, and actors are primarily connected through informal social networks. In generating this theme, we identified several trade-offs related to a small and informally connected stakeholder community, which produced what we

refer to as an inclusivity-exclusivity paradox. Overall, stakeholders shared a positive outlook towards the advantages a small community brings, for example: *"in New Zealand we are extremely fortunate. . . it's that two person connectivity. . . New Zealand is small enough that everyone knows everyone that knows someone"* (W1). A small social network was perceived to increase accessibility and connections between both individuals: *"being smaller is beneficial– easier to get into the right networks compared to larger countries"* (W1), and relevant platforms within institutions: *"we are quite fortunate in that we do have [access to] the government institutions, we do have access to all those agencies. We do have a very flat hierarchy"* (W1).

There was a strong sense of pride regarding NZ's flat hierarchical political structure–particularly when compared to political arrangements globally–and when coupled with the small community factor, NZ was seen to provide fertile grounds for fostering strong, trusting, and enduring interpersonal relationships among policy and research communities. It was emphasised that building strong interpersonal connections was critical for achieving impact at the SPI and linked to this was the view that one's inclusion in Antarctic KE was usually dependent on *"who you know"* (W2). Herein the inclusivity-exclusivity paradox of an informal social network and small community became apparent and as described by one policymaker during workshop 2: *"the strengths of having that close knit community, is also a weakness for inclusivity"* (W2).

Thus, in parallel to the perceived benefits of operating in a small informal social network arrangement, this theme includes a counter narrative regarding the exclusivity of Antarctic KE practices. While the community may be small, stakeholders pointed to a divide between 'insiders' (i.e., those who are actively involved in KE) and 'outsiders' (i.e., those who are, or at least feel, excluded). This led to concerns regarding a lack of diversity within KE, which results in a loss of important voices and perspectives–both of which could serve to enrich KE. In particular, early career researchers (ECRs), social science and humanities scholars and Māori commonly featured as groups that were excluded from Antarctic KE practices. One researcher expressed concerns with respect the impacts that exclusive processes may have on building capacity among the next generation:

> *"Only some people get invited don't they? And it's usually senior scientists. . . So if you don't know the process until you're in it, but you don't get in it until you're a senior scientist. . . Where's the learning for the young scientists?"* (W1).

The informal nature of current KE practices created a lack of transparency regarding what processes and mechanisms currently exist to facilitate KE. This was perceived to create a barrier towards more inclusive engagement, with some referring to the SPI as a *"black box"* (W1). Furthermore, without a clear process for KE, stakeholders found it difficult to understand how inequities among actors were kept in check. For example, researchers that were based near central government in Wellington were perceived as having greater access and visibility to the policy community than those located elsewhere: *"[it is] hard to establish relationships with Wellington when you're not in Wellington"* (W2). Power imbalances among actors were described as limiting the diversity of perspectives engaged in KE and contributing to narrow, technocratic science-policy discussions that are dominated by just a few, generally self-appointed members of a much richer Antarctic community.

## A need for strengthened domestic policy leadership

Stakeholders saw NZ as a strong leader within both Antarctic governance and among the international science community. Notably, the strength and credibility of NZ's Antarctic

research leadership on the international stage was linked to a perception that NZ Antarctic science is relatively independent from policy and politics. The 'objective' 'independence' of NZ Antarctic science was seen to underpin and increase the credibility of NZ's international diplomacy work in the Antarctic policy context:

> "*NZ is relatively well respected on the international stage. We bring good science to the table, I think we're relatively objective and science based in our position*" (W2).

> "*We're actually quite lucky because the research is regarded as relatively independent. In other international institutions, it isn't. The scientists [elsewhere] are representatives of their state and the science reflects that to an extent*" (W1).

Contrary to a sense of strong international leadership, stakeholders identified a lack of domestic coordination and leadership, and expressed a need to strengthen this to achieve more transparent, inclusive and strategic KE. In particular, long-term strategic planning was viewed as critical to the success of Antarctic KE and stakeholders pointed to a leadership deficit on the side of the policy community through remarks such as "*A policy strategy*? *It's just not happening*" (W1); "*There's no kind of direction on where New Zealand wants to go. So it's like, I could come up with a list of things that I think we should do, but is that what the Government wants and what the priorities are*?" (W1); or, "*Unclear understanding of policy strategy, need, networks, communication*" (W1). As expressed by a workshop 2 stakeholder, a major barrier towards improved strategic planning was the blurred connections among a diverse landscape of institutional actors:

> "*We felt there was a lot of overlapping responsibilities in this space and some limited cross-agency institutional discussion to actually clearly establish who has what mandate, and how the people with the mandates are going to actively engage with that group and get the buy-in that they need to be effective*" (W2).

In parallel and as described by one participant, stakeholder entities were perceived to be working in siloes: "*too many separate entities working individually. Hard to know who's who in the zoo*" (W2). A lack of integration across key Antarctic entities was seen as problematic for several reasons. As mentioned, strategic planning becomes difficult and stakeholders agreed that institutional disconnects exacerbated timeline mismatches across policy entities, funders and scientific logistics providers. This was seen to hinder the timely delivery of scientific information:

> "*There needs to be less fragmentation. . . A better line of sight between science and policy, the line of sight is broken. And the agendas are not in-line and that's represented in the timeline as well. The scientists work on long timelines and the policy maker works on reasonably shorter timeline*" (W2).

Inter-agency fragmentation meant that addressing policy needs through the sufficient prioritisation and funding of policy-relevant projects was difficult to achieve. Furthermore, a lack of transparency regarding the connections between key institutional actors was described as compounding the inclusivity-exclusivity paradox above, whereby only those on the 'inside' are clear on what science-policy engagement mechanisms or pathways exist, with 'outsiders' left guessing. Without clear policy leadership, particularly in the strategic planning space, stakeholders shared the understanding that science-policy interactions would continue in an ad hoc, reactive and short-term manner.

Suggestions regarding how to make improvements to domestic policy leadership included further clarifying policy needs and interests along specified timelines, outlining the allocation of investment and resources to meet the needs, assigning leadership roles and responsibilities, and communicating plans to relevant partners and stakeholders. A significant hindrance to making progress, however, was the reported lack of resource on the side of policy. While there was an emphasis on the need for policymakers to clarify their needs to the research community, policymakers at workshop 2 described often feeling short on people-power, with ideas related to improving science-policy dialogues not getting off the ground because *"we'd like to do that, but don't have time"* (W2). Consequently, they cautioned against creating new KE mechanisms or processes, instead emphasising that the focus should be on strengthening and increasing the transparency and diversity of established KE pathways.

## Discussion

Our analysis highlights that the NZ Antarctic community shares a desire to improve the current state of Antarctic science-policy KE practices in several ways. Firstly, there is a strong interest in moving away from linear KE approaches towards more collaborative, participatory and 'co-designed' alternatives. Secondly, there is a need to increase the transparency and inclusivity of current science-policy interactions to legitimise and make space for multiple perspectives within KE. Thirdly, stronger leadership is required from the policy community to facilitate and make clear the connections between Antarctic stakeholder entities so that long-term plans, research programmes and funding mechanisms can strategically address policy requirements and also point to the mechanisms and processes through which researchers and policymakers are able to collaborate. A crosscutting thread shared by breakout groups at both workshops was the need for policy practitioners and researchers to spend more time together to build mutual understandings and enhance capacity on each side of a 'fuzzy' boundary.

Yet, despite a desire to move beyond a linear model, we found that alternative options were poorly demarcated. Firstly, stakeholders held diverse views as to what co-production approaches might look like, and secondly, their descriptions of participatory and collaborative practices were predominantly underpinned by linear repertoires. The former is perhaps unsurprising given the significance of this issue within broader co-production discourse; scholars have criticised the ambiguity surrounding many co-production pursuits and have pointed to weaknesses across an array of conceptualisations as to what constitutes co-production in practice [55,57]. The latter may be partially due to an overrepresentation of the quantitative hypothesis-driven biophysical sciences among the NZ Antarctic research community (as opposed to social science and humanities disciplines), which tend to be underpinned by (post) positivist approaches and thus potentially more linear in their philosophies [84]. As reported elsewhere, it is also likely that this relates to an absence of alternative imaginaries to linear approaches, particularly with respect to balancing asymmetries at the science-policy nexus, such that the policy and research communities are seen as equally responsible for producing knowledge [18].

We found similar asymmetries in our results. For instance, effective science communication was seen as a key competency for achieving successful KE. That is, some actors were perceived as 'doing the science-policy stuff well' if they could clearly communicate scientific findings or translate complex scientific information unidirectionally from science *to* policy. Policymakers were viewed as the passive 'end-users' of credible scientific results, and KE was perceived as the route through which scientific information was *delivered*, rather than a process of knowledge *production* itself. Consequently, the focus was disproportionately on science, moving the narrative away from more participatory practices wherein policymakers and

scientists *jointly* create knowledge through iterative processes. While these perspectives may rightly reflect the current approaches to KE, they also align with claims that have been identified as misguided. For example, they echo three common misperceptions regarding the pathways to achieving actionable knowledge: 1) "science communication is the determining factor in the success of SPIs", 2) "the processes that take place within SPIs facilitate a linear flow of knowledge from scientists to policymakers", and 3) "policymakers are accepting recipients of scientific knowledge" [85:180]. Here we see a missed opportunity with respect to moving beyond simply 'better communicating' the science or 'better articulating' the policy needs for science to address. While it is often the case that policymakers find it difficult to pinpoint their precise knowledge needs [18,22]–particularly in a format that is readily actionable by the scientific community [23]–creating a co-production process through which research and policy communities can *jointly* explore what is meaningful for decision-making could significantly enhance such efforts. Embarking upon such an approach would require co-production competencies among practitioners, but our analysis points to a potential deficit in co-production expertise.

Instead, linear competencies were more evident among Antarctic KE actors, largely serving to reinforce a separation between science and policy worlds [86] and further embed misguided assumptions about the social processes that comprise successful KE [85]. While it is possible that actors invoke linear imaginaries in an attempt to more neatly order this 'fuzzy beast', we wonder to what extent co-production alternatives may have been placed in the too-hard basket. After all, co-production asks actors to legitimise multiple knowledge systems, embrace the deep entanglement of knowledge and social order and embed uncertainty in ways that are meaningful to decision-makers (among other things)–all of which are no small undertakings [87]. It is also plausible that the status quo currently meets the needs of the 'insiders' (at least to a certain extent), leaving any appetite for change with those on the 'outside' who already find themselves disempowered with little influence to create change [58]. In an informal science-policy arrangement that predominantly relies upon person-to-person connections, it is challenging for new voices or perspectives to gain traction when strong and trusting relationships are already well established among inside actors. Any attempts to move away from the current arrangement will require a critical examination of the power relations between actors [85], which may raise uncomfortable and previously unexplored social dynamics. Navigating these sensitivities will require co-production competencies from actors, including being able to question personal assumptions, exercise reflexivity and maintain humility [47,73,88].

Another fundamental step in shifting away from the linear model will be to shine light on how we organise, structure and resource our institutions and to what extent broader systemic issues influence their function with respect to enriching or hindering more collaborative KE practices. This would include carefully considering the systemic and institutional constraints that impede the manoeuvrability of individual practitioners, such that future processes are realistic about the degree to which actors have independence and agency within their organisational confines [18]. As our results suggest, NZ's Antarctic stakeholder entities are poorly integrated, which consequently limits inter-agency interactions and creates a significant barrier in the context of strategic planning. This is of particular concern in the context of implementing the updated *Aotearoa New Zealand Antarctic and Southern Ocean Research Directions and Priorities* document [30]. Set against the backdrop of a fragmented institutional landscape, this strategy currently draws aspirational links between Antarctic research themes and specific policy forums, but does not provide clarity on what processes and connections are in place to ensure these links are enduring and effective in practice. Without establishing *who* or *which* entities are tasked–and thus adequately resourced–to lead on facilitating the necessary dialogues to achieve successful KE (at institutional and individual levels), it is likely that KE

practices will remain as they have been described by the Antarctic stakeholder community: informal, reactive and linear in nature.

Perhaps unsurprisingly, other environmental sectors in NZ reflect similar findings. For instance, the need for greater inclusivity, strategic leadership and clearer policy directives to better align policy priorities with evidence syntheses in NZ's freshwater sector have been identified as imperatives for improving the science-policy dialogues that underpin freshwater policy development [89]. Further exacerbating these challenges is in an overall misalignment between NZ's government strategic priorities on environmental issues and the allocation of research funds, which creates a major barrier at the NZ SPI across a range of environmental issues including those that relate to Antarctica [90–93]. In terms of opportunities to improve the SPI, shifts towards participatory approaches have enabled creative decision-making practices that allow space for multiple knowledge systems in NZ's Integrated Coastal Management sector [63]. Other ATCPs have identified similar challenges (and opportunities) regarding research-policy-strategy mismatches and fragmented domestic Antarctic stakeholder landscapes (e.g., see [39,40]). These findings highlight the importance of gaining greater insights on the domestic drivers and barriers to KE success and sharing lessons across both Antarctic and non-Antarctic contexts at multiple scales.

## A deliberate and careful shift beyond the linear model in NZ

At both workshops, Antarctic stakeholders pointed to a variety of potential opportunities for improving Antarctic science-policy KE practices going forward–many of which have already been identified or explored throughout the literature as pathways to achieving more successful KE [43,65,94–96]. Some suggestions addressed the epistemological and competency barriers that currently hamper more participatory practices, whereas others related to the institutional and systemic challenges.

Epistemological and competency barriers may be overcome by capacity building in both science and policy communities [97]. Ideas included providing mentorship and training opportunities for ECRs to learn about the policy context within which their research might have an impact e.g., through secondment schemes that place ECRs within relevant policy settings [93,96]. Additional workshops like those we ran for the purposes of this study were also seen as effective venues for stimulating discussions among science and policy actors and, if facilitated well, an effective way to build trust and mutual understandings across diverse expertise areas.

Institutional and systemic challenges may be addressed by developing a map of the institutional landscape in NZ to elucidate the connections between key Antarctic stakeholder entities and to clarify the KE mandates, roles and responsibilities related to each. Increasing the transparency of, and accessibility to, current science-policy KE mechanisms was emphasised by our research participants. Other ideas included the development of new incentives for science-policy engagement and rewarding meaningful KE efforts that transcend traditional measures of impact [17,65]. Securing long-term funding for research programmes was also viewed as a way to support the long-term planning required to implement the strategic direction and to help embed science-policy mechanisms within research projects. An important and relevant issue here is that Antarctic domestic policy entities do not always fund research programmes, thus developing strong connections between Antarctic policy entities and funding providers is critical for the production of relevant and useful scientific knowledge [11].

While the above suggestions come with their individual merits, we argue that a more deliberate and careful approach to the Antarctic SPI must be taken to actualise any of their potential benefits. Social scientists have shown that, although our conceptual understandings of KE have

advanced, KE activities in practice are often not theoretically, methodologically or empirically grounded, thus commonly resulting in *ad hoc* KE efforts that fail to achieve their intended outcomes [43,70,98]. With co-production practices rapidly on the rise, scholars have warned practitioners against embarking upon such approaches without first establishing conceptual clarity on their meaning and outcomes, without which co-production is at risk of being framed as a panacea [55,57]. The realities of harnessing pluralism within collaborative practices raises issues of difference, power and conflict; dimensions that may become overshadowed by more aspirational narratives regarding the urgent need for co-production in the face of today's complex problems [85,87]. Thus co-production practices, if not treated carefully, can actually serve to undermine their stated objectives by reinforcing the issues they are attempting to solve [73]. Any transition towards the co-production end of the KE continuum must therefore be deliberate and treated with care.

Our analysis shows that a more deliberate approach would include stronger policy leadership, more equitable, inclusive and diverse KE dialogues and a shift away from linear approaches. This, in turn, would require a dedicated effort to expand co-production competencies and expertise so that theory can better inform practice. We do note a possible caveat, however, with respect to ensuring that KE practices are informed by theory as this assertion comes with linear pretences, i.e., that social-science research examining KE theory and practice must be delivered to, and understood by, practitioners. As discussed, tacit knowledge and know-how play an under-acknowledged role in KE [99] and these knowledge-types do not necessarily materialise through deep theoretical understandings. Rather, they can develop experientially, intuitively or otherwise, which means that co-production practices may come more naturally to some than others without a tangible explanation as to why. While this may be the case, we still see it as important to build theoretical understandings, such that KE practices and practitioners can remain up to date on the most recent learnings from the field.

## Conclusion

Empirical studies that investigate KE in the context of Antarctic governance are in their infancy and our study is the first to situate Antarctic KE practices in NZ within the rapidly advancing field of environmental science and policy studies. We see this as a timely contribution given the unprecedented rate at which Antarctic environments are changing in the Anthropocene [3] and the urgency with which policymakers around the world are required to consider global environmental change as a key input to their decision-making processes. Given the inherently complex nature of KE practices, it is unsurprising that NZ's Antarctic SPI has been likened to a 'fuzzy beast', and we suspect it is likely to remain so. However, through a more deliberate approach from NZ practitioners, we may come closer to 'taming' this beast; not in the sense that we will bring it under predictable or rational control, but rather, set out on a path towards better understanding and working with its complex character. After all, "KE is a flexible process that must be monitored, reflected on and continuously refined" [70:337]. Optimism prevails among Antarctic stakeholders in NZ towards achieving positive change in the science-policy space and is indicative of a capacity and willingness for personal and institutional investments to strengthen Antarctic KE. Our study provides a stepping stone along this iterative journey by highlighting the need for more collaborative, inclusive, diverse and transparent Antarctic science-policy dialogues that are guided by stronger leadership from NZ's Antarctic policy community. Future KE practices might benefit from the co-production of knowledge and action, which requires new ways of working as well as an examination of how institutions influence and govern the relationships between Antarctic decision-making and knowledge production in NZ. This would inevitably raise difficult questions of power and

equity, particularly with regard to whose and what knowledge is legitimate, and who gets to decide. Exercising humility and reflexivity will be critical [47].

Examining and enriching domestic Antarctic KE practices are a matter of local and global significance. Antarctica and the Southern Ocean are integral to the global climate system, home to some of the least impacted ecosystems on Earth and host to an abundance of endemic biodiversity [2,5,100]. But Antarctic environments are changing, and it is primarily the ATCPs that are responsible for ensuring that Antarctic knowledge is mobilised within governance at multiple temporal and spatial scales. This becomes particularly challenging in the face of transboundary issues like climate change that require urgent global solutions and are characterised by high levels of scientific uncertainty, divergent values, conflicts of interest and high stakes policy options [60]. As a multilateral environmental governance institution with evidence-based decision-making at its heart, there is no more urgent time than now to place even greater focus on how the ATCPs individually and collaboratively develop and stabilise their civic (domestic) and intergovernmental (international) epistemologies regarding knowledge production and use [82,101].

Antarctic science-policy practices and scholarship both in NZ and globally would benefit from a research agenda that investigates science-policy KE practices through transdisciplinary and co-production lenses. Chief lines of inquiry could include:

- Comparative analyses of how the different ATCPs understand and arrange KE practices domestically and what bearing these arrangements have on ATS governance more broadly;

- How to define and evaluate KE success at multiple scales, including identifying Antarctic science-policy 'bright spots' [41];

- How science-policy practices are stabilized across the different ATS policy bodies;

- What repertoires of knowledge brokering are invoked in which Antarctic policy contexts and why;

- How might co-production practices be facilitated at national and international scales;

- What lessons from other environmental governance contexts (in NZ and globally) could be drawn on to enrich Antarctic KE;

- What is the role of non-state and non-human actors in science-policy processes; and,

- What is role of trust at the Antarctic SPI in an increasingly complex and polarized world.

The Antarctic community should dedicate greater energy and efforts towards disentangling the complexities of Antarctic science-policy discourse through explorative, participatory methods that engage researchers, policymakers and other key actors in the co-production of Antarctic knowledge and governance.

## Supporting information

**S1 File. Detailed workshop methods.**
(DOCX)

## Acknowledgments

A huge thank you to all of the workshop participants and our colleagues who volunteered as breakout group facilitators at workshop 1. Special thanks to Timo Maas for the comments that he kindly provided on a previous version of this manuscript. Thanks to Seth Skyora-Bodie for

his advice on methods in the early stages of workshop planning. Thank you Antarctica New Zealand for including workshop 1 in the 2021 NZ Antarctic Science Conference. We are grateful for the helpful feedback provided by the anonymous reviewers of this paper, which served to enrich its quality. This paper contributes to the '*Integrated Science to Inform Antarctic and Southern Ocean Conservation*' (Ant-ICON) Scientific Research Programme of the Scientific Committee on Antarctic Research (SCAR), in particular its Synthesis Theme 1.

## Author Contributions

**Conceptualization:** Natasha Blaize Gardiner, Neil Gilbert, Daniela Liggett.

**Data curation:** Natasha Blaize Gardiner.

**Formal analysis:** Natasha Blaize Gardiner.

**Funding acquisition:** Neil Gilbert, Daniela Liggett.

**Investigation:** Natasha Blaize Gardiner, Neil Gilbert, Daniela Liggett.

**Methodology:** Natasha Blaize Gardiner, Neil Gilbert, Daniela Liggett.

**Project administration:** Natasha Blaize Gardiner.

**Supervision:** Neil Gilbert, Daniela Liggett.

**Visualization:** Natasha Blaize Gardiner, Neil Gilbert, Daniela Liggett.

**Writing – original draft:** Natasha Blaize Gardiner.

**Writing – review & editing:** Natasha Blaize Gardiner, Neil Gilbert, Daniela Liggett.

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
