## [Decision Letter · Decision Letter 0]

5 Sep 2023

PONE-D-23-12844Taming a ‘fuzzy beast’? Stakeholder perspectives on Antarctic science-policy knowledge exchange practices in New ZealandPLOS ONE

Dear Dr. Gardiner,

Thank you for submitting your manuscript to PLOS ONE. After careful consideration, we feel that it has merit but does not fully meet PLOS ONE’s publication criteria as it currently stands. Therefore, we invite you to submit a revised version of the manuscript that addresses the points raised during the review process.

We look forward to receiving your revised manuscript.

Kind regards,

Daniel de Paiva Silva, Ph.D.

Academic Editor

PLOS ONE

“A huge thank you to all of the workshop participants and our colleagues who volunteered as breakout group facilitators at workshop 1. Special thanks to Timo Maas for the comments that he kindly provided on a previous version of this manuscript. Thanks to Seth Skyora-Bodie for his advice on methods in the early stages of workshop planning. Thank you Antarctica New Zealand for including workshop 1 in the 2021 NZ Antarctic Science Conference. We are grateful for the helpful feedback provided by the anonymous reviewers of this paper, which served to enrich its quality. We acknowledge and thank the Antarctic Science Platform for funding this project through grant ASP-023-4. This paper contributes to the ‘Integrated Science to Inform Antarctic and Southern Ocean Conservation’ (Ant-ICON) Scientific Research Programme of the Scientific Committee on Antarctic Research (SCAR), in particular its Synthesis Theme 1.”

“NBG, NG and DL received funding for this research from the Antarctic Science Platform through grant ASP-023-4. The funders had no role in study design, data collection and analysis, decision to publish, or preparation of the manuscript.

https://www.antarcticscienceplatform.org.nz/”

Additional Editor Comments:

Dear Dr. Gardiner,

Please note that I have recently assumed this manuscript as an editor. When I assumed the editorship of this study, both reviewers had already provided their decisions. This is the reason of this decision letter in such a short period from the last message you sent to the journal.

After this review round, both reviewers believe your manuscript has merits and deserves to be published in PLoS One. Therefore, please consider all issues raised nby both reviewers, improve your text, and resubmit it by November 5th, 2023 along with a rebuttal letter in which you will show all changes that were made and those you disregarded. Please do not forget to explain why you did not considered the reviewers' suggestions in these latter cases.

Sincerely,

Daniel Silva, PhD

Reviewers' comments:

Reviewer's Responses to Questions

**Comments to the Author**

1. Is the manuscript technically sound, and do the data support the conclusions?

Reviewer #1: Yes

Reviewer #2: Yes

2. Has the statistical analysis been performed appropriately and rigorously? 

Reviewer #1: N/A

Reviewer #2: N/A

3. Have the authors made all data underlying the findings in their manuscript fully available?

Reviewer #1: No

Reviewer #2: No

4. Is the manuscript presented in an intelligible fashion and written in standard English?

Reviewer #1: Yes

Reviewer #2: Yes

5. Review Comments to the Author

Reviewer #1: Taming a ‘fuzzy beast’? Stakeholder perspectives on Antarctic science-policy knowledge exchange practices in New Zealand

This is a timely and very well written study. However, I find the focus solely on NZ and the lack of a recognition of the relevance of this work to broader governance of Antarctica problematic. I propose a few changes will which improve the credibility and the broader relevance of the study.

Introduction

Currently, it is not clear in the introduction why anyone outside of NZ might be interested in reading the study. The authors have noted the need for engagement in multi-directional knowledge exchange in Antarctic decision-making more broadly (lines 6-7), so why only focus on calls for this in NZ (lines 17-25). I understand that the focus of the study is NZ, but surely this has significance for all ATS participants? The authors would show the broader benefit of this study by also including some context from other relevant countries here. Also, has the Antarctic science-policy interface been studied anywhere (lines 26-33)?

Conceptual framework

It is not immediately clear what the purpose of the conceptual framework is? Normally it would be a representation of the relationship expected between variables, or the characteristics or properties to be studied. Yet it doesn’t directly link to the research questions as the authors do not appear to be asking where on the continuum NZ Antarctic KE lies. Some further clarification about what role this plays (other than just context) would be helpful for the reader.

Methods

The limitations of the methodology should be presented in the main text of the document. It is unlikely that a reader will go to the supplementary material to read this and not including them in the main body risks readers being unable to fully discern the credibility of the study's conclusion.

Discussion

The discussion currently lacks a clear explanation of whether the desired changes to the KE process identified in the results has been found in other ATS partner states, or even in environmental decision-making more broadly (lines 360-370). I would imagine that some of these findings are not particular to just this case example. Identifying whether these are broader issues would make the findings of the study of broader interest.

Reviewer #2: Overall, I don't have any major concerns in regards to this paper. It is well written and outlines the research, frameworks, and outcomes well. I commend you on your research. I do have a few smaller questions/comments.

This are more in the vein of additional context required/more info/have you thought of this?

* For all quotes, there are some codes(?) at the end (such as PL W2/BC3 W1). Im guessing they mean Workshop 1 (W1) and Workshop 2 (W2) but the other parts are more confusing. Are they really required? I am not 100% sure they are.

* More a comment - a lot of this is what I would call 'stated preference' where the scientists and policy makers are stating what they think is needed. While still valid, I think that its important to note that scientists and policy makers might not actually choose this approach (revealed preference) if given the opportunity.

* Page 15, Line 296: While Wellington is the home of MFAT and Central Govt, I am surprised the Christchurch-centric framing was not also highlighted since Antarctica NZ is based there.

* Page 17, Lines 353-358: I think this is potentially a wasted opportunity as this section seemed to gloss over the CCAMLR successes and not acknowledge the details and thoughts of the community in relation to CCAMLR, particularly from Workshop 1?

* Page 22, End of Conclusion section: Nothing really pressing here, but I was thinking "What is the key aspect that you would want a ready to take away from reading this article". I feel that this conclusion, was a bit brief and there could be more clarity specifically around key messages or aspects for the future of the SPI and KE within the NZ Antarctic science-policy community.

* References 54 and 55: These two seem formatted with an indent. I don't think this is an issue but just wanted to highlight if it was.

* Supplementary Information - Appendix 2, Pages 8 and 9: A couple of referencing errors here.

6. PLOS authors have the option to publish the peer review history of their article (what does this mean?). If published, this will include your full peer review and any attached files.

Reviewer #1: No

Reviewer #2: No

---

## [Author Response · Author response to Decision Letter 0]

22 Sep 2023

We have provided our responses to the reviewers in the attached file "response to reviewers", however, we also include them below. This is a timely and very well written study. However, I find the focus solely on NZ and the lack of a recognition of the relevance of this work to broader governance of Antarctica problematic. I propose a few changes will which improve the credibility and the broader relevance of the study. 

Thank you for this feedback. This paper is comprises a PhD thesis chapter and the NZ study that is reported on in the paper is situated within the broader context of Antarctic governance elsewhere in the thesis, hence the omission of these details from this particular manuscript. However, we agree that the article in its present form is somewhat NZ-centric & could be broadened to consider a more global Antarctic governance lens to appeal to an international audience. To address this request, we have made several changes throughout the manuscript, which we document below. Thank you for this suggestion!

Introduction

Currently, it is not clear in the introduction why anyone outside of NZ might be interested in reading the study. The authors have noted the need for engagement in multi-directional knowledge exchange in Antarctic decision-making more broadly (lines 6-7), so why only focus on calls for this in NZ (lines 17-25). I understand that the focus of the study is NZ, but surely this has significance for all ATS participants? The authors would show the broader benefit of this study by also including some context from other relevant countries here. Also, has the Antarctic science-policy interface been studied anywhere (lines 26-33)?

Thank you, this is a good point & we have now added new text to the abstract, introduction, discussion and conclusion sections that helps to bring the study into a broader ATS/global lens. 

Introduction – see lines 16 – 38 on the clean manuscript that does not include track changes. Also see new text lines 49 – 61 (clean ms). 

Discussion – see new paragraph lines 478 – 491 (clean ms). 

Conclusion – see new paragraphs lines 564 – 595 (clean ms). 

Conceptual framework

It is not immediately clear what the purpose of the conceptual framework is? Normally it would be a representation of the relationship expected between variables, or the characteristics or properties to be studied. Yet it doesn’t directly link to the research questions as the authors do not appear to be asking where on the continuum NZ Antarctic KE lies. Some further clarification about what role this plays (other than just context) would be helpful for the reader.

Thank you for your comments regarding the conceptual framework. The purpose of providing a CF in this case is to help situate the findings of our study within wider science-policy theory, i.e., the CF helps to situate the study in its theoretical, conceptual, axiological and practical context. Qualitative research is an iterative process whereby conceptual understandings grow alongside the research process (Ravitch & Riggan, 2016). It is not common that a qualitative researcher embarks upon a project with hypotheses or expectations (as you have suggested) in mind, nor a predetermined conceptual backdrop within which findings will neatly fit. As is the case with most qualitative approaches, our understandings of NZ KE developed alongside our CF in an iterative, nonlinear fashion. Early in our analysis, for example, we recognised the linear ways in which NZ practitioners understood and described Antarctic KE, which prompted a more in-depth exploration of the co-production literature to understand the theoretical assumptions underpinning linear approaches. It was through this line of enquiry that we began developing the KE continuum, which is the central focus of our first theme: “A desire to move beyond the linear model”. 

Furthermore, our study adopts a reflexive TA approach, which was developed – and continues to be updated – by psychology scholars Virginia Braun and Victoria Clarke (see their seminal paper on TA here: Using thematic analysis in psychology: Qualitative Research in Psychology: Vol 3, No 2 (tandfonline.com)). Braun and Clarke (2022), argue that while keeping your research question in mind during thematic development is important, “it does not mean looking for a direct answer to that research question… It means instead generally keeping in mind what your interest in the topic (dataset) is, and exploring patternings that might illuminate our understating of the issue” (p. 90). Our research comprises two parts: “how does the Antarctic science-policy interface(s) function in NZ?” and “what are the drivers and barriers for success?” We have clearly identified that KE in NZ tends to be situated towards the linear end of the KE continuum, which we believe addresses part 1 of the research question regarding function. Regarding part 2 of the research question (drivers and barriers), we have shown that the linear model appears to be perceived as a barrier & that a shift beyond the linear model towards more collaborative practices would help to drive more successful KE. Through the development of the CF, we were therefore able to enrich our understandings of the dataset and our ability to communicate our findings. We also note that most of the Antarctic science-policy literature to date has been very weak in theory, which means that our study provides both a theoretical and empirical contribution to the Antarctic social science literature more broadly. 

We point the reviewer to a couple of examples that show how conceptual frameworks are used to inform the analysis: 

Policy-makers perspectives on credibility, relevance and legitimacy (CRELE) - ScienceDirect

Maas co-producing science policy - Google Scholar

Protecting Antarctica through Co-production of actionable science: Lessons from the CCAMLR marine protected area process - ScienceDirect

For the above reasons, and because providing information about the CF we adopt is essential from a methodological and contextual perspective, we have not significantly change the section about the CF, except for making minor changes that help clarify its purpose in lines 81 – 83 in the clean ms. 

Braun, V., & Clarke, V. (2006). Using thematic analysis in psychology. Qualitative research in psychology, 3(2), 77-101.

Braun, V., & Clarke, V. (2022). Thematic analysis: A practical guide. Sage Publications. 

Ravitch, S. M., & Riggan, M. (2016). Reason & rigor: How conceptual frameworks guide research. Sage Publications.

Methods

The limitations of the methodology should be presented in the main text of the document. It is unlikely that a reader will go to the supplementary material to read this and not including them in the main body risks readers being unable to fully discern the credibility of the study's conclusion.

We agree with this observation and have now added the limitations to the main text and thank the reviewers for this excellent suggestion. We have also made some minor adjustments to the text. 

See lines 207 – 223 (clean ms). 

Discussion

The discussion currently lacks a clear explanation of whether the desired changes to the KE process identified in the results has been found in other ATS partner states, or even in environmental decision-making more broadly (lines 360-370). I would imagine that some of these findings are not particular to just this case example. Identifying whether these are broader issues would make the findings of the study of broader interest.

Thank you. We have now added the following:

- New text in the discussion section (lines 487 – 491 in the clean ms) that points to a couple of relevant reports on KE in the domestic settings in Australia and Chile. 

- We have added new paragraphs to the conclusion (lines 564 – 595 clean ms) to bring the global perspective back in and create an arc from the beginning of the paper to the end. 

At many points throughout the results & discussion we state (or now state) that our findings have been reflected elsewhere in the science-policy literature thereby situating our findings in a broader context (as you have suggested), therefore we have not added a whole lot more context in this regard:

Lines 236 – 237 (clean ms): “While these findings are similar to recent studies that explore KE models in practice…”

Lines 414 – 417 (clean ms): “The former is perhaps unsurprising given the significance of this issue within broader co-production discourse; scholars have criticised the ambiguity surrounding many co-production pursuits and have pointed to weaknesses across an array of conceptualisations as to what constitutes co-production in practice”

Lines 421 – 423 (clean ms): “As reported elsewhere, it is also likely that this may relate to an absence of alternative imaginaries to linear approaches, particularly with respect to balancing asymmetries at the science-policy nexus, such that the policy and research communities are seen as equally responsible for producing knowledge (Maas et al., 2022). 

We found similar asymmetries in our results.”

Lines 432 – 437 (clean ms): “While these perspectives may rightly reflect the current approaches to KE, they also align with claims that have been identified as misguided. For example, they echo three common misperceptions regarding the pathways to achieving actionable knowledge: 1) “the processes that take place within SPIs facilitate a linear flow of knowledge from scientists to policymakers”, 2) “science communication is the determining factor in the success of SPIs”, and 3) “policymakers are accepting recipients of scientific knowledge”. (Jagannathan et al., 2023).

Lines 493 – 495 (clean ms): “At both workshops, Antarctic stakeholders pointed to a variety of potential opportunities for improving Antarctic science-policy KE practices going forward – many of which have already been identified or explored throughout the literature as pathways to achieving more successful KE”

Overall, I don't have any major concerns in regards to this paper. It is well written and outlines the research, frameworks, and outcomes well. I commend you on your research. I do have a few smaller questions/comments.

Thank you for this very kind feedback. 

For all quotes, there are some codes(?) at the end (such as PL W2/BC3 W1). Im guessing they mean Workshop 1 (W1) and Workshop 2 (W2) but the other parts are more confusing. Are they really required? I am not 100% sure they are.

Thank you, we have now changed all codes to W1 or W2 to represent which workshop each quote came from. 

More a comment - a lot of this is what I would call 'stated preference' where the scientists and policy makers are stating what they think is needed. While still valid, I think that its important to note that scientists and policy makers might not actually choose this approach (revealed preference) if given the opportunity.

This is an excellent point and if PLoS One offered footnotes, we would have liked to have captured this observation in a footnote. We are not sure how best to fit this into the main text and have therefore taken no further action. 

Page 15, Line 296: While Wellington is the home of MFAT and Central Govt, I am surprised the Christchurch-centric framing was not also highlighted since Antarctica NZ is based there.

Thank you for this very astute comment. Interestingly, the ‘Wellington effect’ was not described in the context of Christchurch by any of our participants. We speculate that this might be because many people in the NZ Antarctic community incorrectly perceive Antarctica New Zealand to be mainly a logistics provider. The policy work undertaken by Antarctica New Zealand seems to have less visibility among the community. 

Page 17, Lines 353-358: I think this is potentially a wasted opportunity as this section seemed to gloss over the CCAMLR successes and not acknowledge the details and thoughts of the community in relation to CCAMLR, particularly from Workshop 1?

This is a really good point. We have now removed this section and have added a few supplementary lines in the limitations section to clarify that we are not specifically focused on the CCAMLR or ATCM/CEP policy context. Unfortunately, we did not have time to explore knowledge exchange in the specific context of CCAMLR or the ATCM/CEP at the workshops. It would be interesting to see a future research project that takes a comparative approach towards NZ’s CCAMLR vs ATCM/CEP science-policy work. One major point of difference are the funding mechanisms for policy-relevant CCAMLR research versus policy-relevant ATCM/CEP. The latter being far less formalised. 

Page 22, End of Conclusion section: Nothing really pressing here, but I was thinking "What is the key aspect that you would want a ready to take away from reading this article". I feel that this conclusion, was a bit brief and there could be more clarity specifically around key messages or aspects for the future of the SPI and KE within the NZ Antarctic science-policy community.

Thank you for this useful reminder. We have now adjusted the introductory and concluding sections with the aim to provide a more international/ATS lens as per the comments of the 1st reviewer. We hope this has strengthened the key messages in the conclusion both with respect to future KE in NZ and beyond. 

References 54 and 55: These two seem formatted with an indent. I don't think this is an issue but just wanted to highlight if it was.

Thank you. We have since reformatted all of the references to match the PloS One style. 

Supplementary Information - Appendix 2, Pages 8 and 9: A couple of referencing errors here.

We may need further clarification on the exact nature of the referencing errors as were unable to find any referencing errors on pages 8 and 9 of our S1 File.

---

## [Decision Letter · Decision Letter 1]

25 Oct 2023

Taming a ‘fuzzy beast’? Stakeholder perspectives on Antarctic science-policy knowledge exchange practices in New Zealand

PONE-D-23-12844R1

Dear Dr. Gardiner,

We’re pleased to inform you that your manuscript has been judged scientifically suitable for publication and will be formally accepted for publication once it meets all outstanding technical requirements.

Kind regards,

Daniel de Paiva Silva, Ph.D.

Academic Editor

PLOS ONE

Reviewers' comments:

Reviewer's Responses to Questions

**Comments to the Author**

1. If the authors have adequately addressed your comments raised in a previous round of review and you feel that this manuscript is now acceptable for publication, you may indicate that here to bypass the “Comments to the Author” section, enter your conflict of interest statement in the “Confidential to Editor” section, and submit your "Accept" recommendation.

Reviewer #2: All comments have been addressed

2. Is the manuscript technically sound, and do the data support the conclusions?

Reviewer #2: (No Response)

3. Has the statistical analysis been performed appropriately and rigorously? 

Reviewer #2: (No Response)

4. Have the authors made all data underlying the findings in their manuscript fully available?

Reviewer #2: (No Response)

5. Is the manuscript presented in an intelligible fashion and written in standard English?

Reviewer #2: (No Response)

6. Review Comments to the Author

Reviewer #2: (No Response)

7. PLOS authors have the option to publish the peer review history of their article (what does this mean?). If published, this will include your full peer review and any attached files.

Reviewer #2: No

---

## [Editor Report · Acceptance letter]

14 Nov 2023

PONE-D-23-12844R1 

Taming a ‘fuzzy beast’? Stakeholder perspectives on Antarctic science-policy knowledge exchange practices in New Zealand 

Dear Dr. Gardiner:

I'm pleased to inform you that your manuscript has been deemed suitable for publication in PLOS ONE. Congratulations! Your manuscript is now with our production department. 

Kind regards, 

on behalf of

Dr. Daniel de Paiva Silva 

Academic Editor

PLOS ONE